# The ERAD Pathway Participates in Fungal Growth and Cellulase Secretion in *Trichoderma reesei*

**DOI:** 10.3390/jof9010074

**Published:** 2023-01-04

**Authors:** Cheng Yao, Mengjie Yan, Kehang Li, Weihao Gao, Xihai Li, Jiaxin Zhang, Hong Liu, Yaohua Zhong

**Affiliations:** State Key Laboratory of Microbial Technology, Institute of Microbial Technology, Shandong University, Qingdao 266237, China

**Keywords:** *Trichoderma reesei*, ERAD, fungal growth, cellulase secretion, endoplasmic reticulum stress

## Abstract

*Trichoderma reesei* is a powerful fungal cell factory for the production of cellulolytic enzymes due to its outstanding protein secretion capacity. Endoplasmic reticulum-associated degradation (ERAD) plays an integral role in protein secretion that responds to secretion pressure and removes misfolded proteins. However, the role of ERAD in fungal growth and endogenous protein secretion, particularly cellulase secretion, remains poorly understood in *T. reesei*. Here, we investigated the ability of *T. reesei* to grow under different stresses and to secrete cellulases by disrupting three major genes (*hrd1*, *hrd3* and *der1*) involved in the critical parts of the ERAD pathway. Under the ER stress induced by high concentrations of DTT, knockout of *hrd1*, *hrd3* and *der1* resulted in severely impaired growth, and the mutants Δ*hrd1* and Δ*hrd3* exhibited high sensitivity to the cell wall-disturbing agents, CFW and CR. In addition, the absence of either *hrd3* or *der1* led to the decreased heat tolerance of this fungus. These mutants showed significant differences in the secretion of cellulases compared to the parental strain QM9414. During fermentation, the secretion of endoglucanase in the mutants was essentially consistent with that of the parental strain, while cellobiohydrolase and β-glucosidase were declined. It was further discovered that the transcription levels of the endoglucanase-encoding genes (*eg1* and *eg2*) and the cellobiohydrolase-encoding gene (*cbh1*) were not remarkedly changed. However, the β-glucosidase-encoding gene (*bgl1*) was significantly downregulated in the ERAD-deficient mutants, which was presumably due to the activation of a proposed feedback mechanism, repression under secretion stress (RESS). Taken together, our results indicate that a defective ERAD pathway negatively affects fungal growth and cellulase secretion, which provides a novel insight into the cellulase secretion mechanism in *T. reesei*.

## 1. Introduction

Filamentous fungi have the natural characteristics of effectively degrading lignocellulosic materials and have been widely used in cellulase production for many years, among which *Trichoderma reesei* stands out for its excellent cellulase secretion ability and is used as a model industrial fungus [1,2]. Endoglucanases (EGs, EC3.2.1.4), cellobiohydrolases (CBHs, EC 3.2.1.91) and β-glucosidase (exactly BGL1, EC 3.2.1.21) are the major parts of the cellulose-degrading enzyme system in the fungal secretome [3]. With the growing demand for cellulases in industry, many attempts have been undertaken to enhance the production of cellulases, such as optimizing enzyme composition by introducing one or more cellulase genes through genetic engineering means, regulating cellulase expression through transcription factor strategies, etc. [4,5]. With the completion of chromosome sequencing, *T. reesei* has been proven to have a brilliant ability to secrete proteins, especially cellulase. The processes of cellulase biosynthesis and secretion *in T. reesei* is as follows [6]: (I) Cellulase genes are transcribed and then translated by ribosomes into polypeptides. The transcription of cellulase genes is influenced by many factors, the most important of which is the regulation of transcription factors, such as Xyr1, Cre1, Ace2, Ace3, BglR, etc. (II) Efficient protein processing and quality control in the ER. The key to cellulase secretion lies in the correct folding and glycosylation modification of secreted proteins in the ER to mature conformation, as well as the removal of incorrect or insufficient folding by the protein control system to maintain ER homeostasis. (III) Effective vesicle trafficking contributes to cellulase secretion. Vesicle trafficking helps the secreted cellulase from the ER to the Golgi apparatus and finally to the extracellular matrix. (IV) The possible degradation of the secreted cellulase in the vacuole. The secreted cellulase in some cases could be directed to the vacuole and degraded such as the disordered protein transport network. In particular, it was reported that the ER is vital for improving the secretion capacity in fungi [7]. However, the mechanism of protein secretion in the ER of *T. reesei* has not been clearly explored.

In eukaryotes, the ER is the entrance to the secretory pathway and provides the intracellular environment for protein folding and maturation. The newly synthesized polypeptides are co-translationally transported across the ER membrane via the ribosome-Sec61 translocation mechanism [8]. The ER has a stringent quality control system, including unfolded protein response (UPR) and ER-associated degradation (ERAD) [9,10]. When the accumulation of large amounts of unfolded or misfolded proteins results in ER stress, cells initiate UPR to monitor protein folding levels and readjust the folding capacity to match the synthetic load, and then they induce the expression of downstream protein folding and processing-related genes, such as disulfide isomerase gene *pdi1*, molecular chaperone gene *bip1*, etc. [11,12]. Therefore, *bip1* and *pdi1* are frequently used as target genes to report UPR levels. ERAD is responsible for the degradation of misfolded and even normal proteins, limiting the cellular stress and cytotoxicity caused by the accumulation of proteins to ensure protein homeostasis [13]. When the abnormal proteins fail to possess the proper conformation, they are translocated across the ER membrane into the cytoplasm, where they are degraded by the 26S proteasome [14]. A central component of this pathway is the E3 ubiquitin ligase Hrd1, which mainly mediates the degradation of folding-defective proteins in the ER lumen [15]. Meanwhile, the ligase Hrd1 and three other membrane proteins (Hrd3, Der1, Usa1) form a retrograde translocation channel [16]. Hrd3 is a single transmembrane protein with a large lumen domain that interacts with substrates and Hrd1 [17]. Der1 is involved in targeting ERAD substrates for cytoplasmic degradation as an essential transmembrane element [18]. In *Saccharomyces cerevisiae*, Hrd3p was found to regulate Hrd1p activity and stability, while Der1p was shown to be indispensable for the degradation of membrane-bound substrates [16,19].

Several steps that occur in the ER secretory pathway of fungi have been identified as potential bottlenecks in protein production [20]. For instance, engineered ER folding and glycosylation pathways have been reported to enhance cellulase production in *T. reesei* [21]. In addition, it was found that absence of the ERAD transmembrane protein Der1 resulted in activation of the UPR and increased the secretion of a heterologous endoglucanase in *S. cerevisiae* [22,23]. Lack of another ERAD-related factor DoaA could positively affect the secretion of the heterologous β-glucuronidase in *A. niger* [24]. Especially, deletion of *derA* in *A. niger* could promote production of the heterologous fusion protein, glucoamylase–glucuronidase [25]. However, the functional roles of the key ERAD components in endogenous protein secretion in fungi remain rarely known.

This study aimed to explore the roles of three key ERAD components (Hrd1, Hrd3 and Der1) in the ERAD pathway in *T. reesei*. The three functional genes encoding different parts of the ERAD pathway were disrupted, and the resulting mutants had different degrees of defects in their growth under ER stress, cell wall perturbation, and heat stress. In addition, the deficient ERAD pathway reduced the secretion of β-glucosidase and cellobiohydrolase but had no effect on the secretion of endoglucanase. Transcriptional level analysis revealed that the decrease in β-glucosidase secretion was due to the reduced transcription of the β-glucosidase-coding gene *bgl1*. Our results indicated that an intact ERAD pathway is required for normal fungal growth under stresses and cellulase secretion during fungal fermentation.

## 2. Materials and Methods

### 2.1. Fungal Strains and Culture Conditions

*Trichoderma reesei* QM9414 (ATCC 26921) was used as the host strain for gene knockout. Strains were routinely cultured on minimal medium (MM) for vegetative growth [26]. In order to prepare spore suspension, all fungal strains were maintained on potato dextrose agar (PDA) medium containing peeled potato extract 200 g/L, glucose 20 g/L and agar 20 g/L at 30 °C for 5–7 days. Fresh spores were spread on the PDA plate covered with cellophane, and then, the hyphae were lysed to harvest protoplasts for transformation [26]. The esculin plate was used to detect the ability of the strains to secret β-glucosidase, since β-glucosidase would hydrolyze the substrate esculin to glucose and esculetin that could bind with ferric ions to yield a dark-brown color in the agar around the fungal colonies. The plate medium contained 3 g/L esculin, 0.5 g/L ferric citrate, 10 g/L sodium carboxymethyl cellulose (CMC–Na), and 20 g/L agar. The supernatant liquid and mycelium used for enzyme activity determination and RNA extraction were obtained by shake-flask fermentation. The spore suspension was adjusted to 10^8^ conidia/mL using a hemocytometer for counting and inoculated into 50 mL seed culture medium, cultured at 200 rpm and 30 °C for 30–36 h. Afterward, 10 mL of the pre-culture was transferred into 100 mL of the cellulase production medium (CPM) [27] and grown at 200 rpm and 30 °C. The fermentation supernatant was taken every 24 h for cellulase production test. The mycelium was collected after 36 h of growth for RNA extraction. The seed culture medium was composed of 10 g/L glucose, 5 g/L (NH_4_)_2_SO_4_, 15 g/L KH_2_PO_4_, 0.6 g/L CaCl_2_·2H_2_O, 0.5 g/L MgSO_4_·7H_2_O and 2 g/L peptone [27]. The components of CPM were as follows: 20 g/L microcrystalline cellulose, 15 g/L KH_2_PO_4_, 0.6 g/L CaCl_2_·2H_2_O, 0.5 g/L MgSO_4_·7H_2_O and 5 g/L (NH_4_)_2_SO_4_ or 20 g/L corn steep steam (CSL) [27].

### 2.2. Construction of the ERAD-Related Gene Deletion Strains

The ERAD-related gene deletion cassettes were constructed by double-joint PCR technology [28]. Two or more gene fragments could be connected in vitro by means of terminal complementarity and overlapping extension. The deletion mutants were constructed using a classic method based on homologous recombination of deletion cassettes. Phanta^®^ Super-Fidelity DNA Polymerase (Vazyme Biotech Co., Ltd., Nanjing, China) was used for PCR amplification, and DNA fragments were purified using Gel Extraction Kit (Omega, Norcross, GA, USA). Primer synthesis and DNA sequencing were provided by Sangon Inc (Shanghai, China). Primer sequences used in this article are indicated in Appendix A.

To obtain the ERAD-related gene deletion strain, the disruption construct containing the hygromycin resistance gene *hph* and homologous flanking regions of the target gene was carried out. The 5′- and 3′-flanking regions were generated from *T. reesei* QM9414 genome using the corresponding primer pairs, *hrd1*-UF/ *hrd1*-UR, *hrd1*-DF/ *hrd1*-DR, *hrd3*-UF/ *hrd3*-UR, *hrd3*-DF/ *hrd3*-DR, *der1*-UF/*der1*-UR, and *der1*-DF/*der1*-DR, respectively. According to the plasmid pAN7-1, the primer pair *hph*-F/*hph*-R was designed to amplify the *hph* gene. Three single fragments were fused without primers, and then, the nest primers were used to create gene replacement cassettes (Appendix A). The knockout cassette of each gene was verified to contain upstream and downstream homologous arms as well as selective marker gene *hph* (Appendix A). Ultimately, the knockout cassettes were separately transformed into the protoplasts to acquire the ERAD-related gene deletion strains, essentially as previously described [26]. The candidate transformants were directly screened on MM with 300 µg/mL hygromycin B. The transformants were purified, and their stability (the stability of the resistance gene *hph* in the genome and the knockout transformants) was determined by using MM plate with 0.1% Triton X-100 and 300 µg/mL hygromycin B to propagate three times in succession according to the method of Gruber et al. [29]. To verify the integration of the deletion cassette at the correct locus, transformants were analyzed by colony PCR using genomic DNA from resistant colonies as templates, thereby obtaining the ERAD-defective mutants that contained upstream and downstream anchoring as well as internal gene knockout (Appendix A). In order to further verify that the ERAD-related gene in the ERAD-defective mutants was knocked out, the RT-qPCR result clearly showed that the *hrd1*/*hrd3*/*der1* gene cannot be expressed in the corresponding knockout strain (Δ*hrd1*/Δ*hrd3*/Δ*der1*) (Appendix A). The ERAD-defective mutants with the corresponding gene deletion were named as Δ*hrd1*, Δ*hrd3* and Δ*der1*, respectively.

### 2.3. Phenotypic Analysis of Knockout Strains under Pressure

For plate growth assays under pressure, MM solidified by the addition of 2% agar was used (as described above). For the purpose of studying the sensitivity of ERAD deletions under endoplasmic reticulum pressure, radial extension rates of the ERAD mutants were determined by inoculating mycelium blocks with the same density on MM supplemented with 1–15 mM DTT, and growth status at 30 °C was followed for 3 days. Then, 10 mM DTT was often added to the medium inducing the ER stress in *T. reesei* [30]. In this study, we used a wider range of DTT concentrations (1–15 mM) to finely explore the ER stress resistance in the knockout strains. The spores of *T. reesei* QM9414 were treated with the proteasome inhibitor MG132 (20 or 40 μM) as the positive control. The concentration of MG132 was used as described by Wang et al. [31]. To explore the effect of the absence of ERAD components on cell wall stability, spores with the concentration of 10^5^/mL were spotted on MM with different concentrations of CFW (0, 20, 40, 80 μg/mL) or CR (0, 20, 40, 80 μg/mL) (both containing 10 mM DTT), cultivated at 30 °C and monitored for 3 days. The concentrations of CFW or CR referred to the method of Ram et al., the concentration gradients of 0, 20, 40 and 80 μg/mL were set for the study [32]. For the sake of investigating the heat stress tolerance of the ERAD mutants, approximately 10^5^ conidiospores of the ERAD mutants were dropped on the center of the MM plates, which were developed at 25, 30 and 37 °C for 3 days prior to examination. Three parallel plate experiments ensured the accuracy of the results.

### 2.4. Cellulase Activity Assays

For analysis of cellulase secretion capabilities, the filter paper activity (FPA), endoglucanase (EG) activity, cellobiohydrolase (CBH) activity, and β-glucosidase (BGL) activity were determined in the culture supernatants after fermentation. By using Whatman No.1 filter paper (Sigma-Aldrich) as a substrate to measure FPA activity, the experimental details such as the reaction system and temperature were as described before [33]. Endoglucanase activity was determined by using soluble CMC-Na substrate combined with the classic DNS method, and β-glucosidase activity was analyzed by using p-nitrophenyl-β-D-glucoside (pNPG) as the substrate according to an optimized method based on previous research [34]. Cellobiohydrolase activity was assayed with p-nitrophenyl-β-D-cellobioside (pNPC) as the substrate [35]. One enzyme activity unit was defined as the amount of enzyme required to release 1 μmol reducing sugar (FPA, EG) or p-nitrophenol (CBH, BGL1) per minute under the measurement conditions. The fermentation liquor of *T. reesei* QM9414 was used as a blank control. At least three parallel fermentations were carried out, and the enzyme activity was repeatedly measured three times in each fermentation to select the most representative result.

### 2.5. Total RNA Extraction and Real-Time Fluorescence Quantitative PCR

For total RNA extraction, fungal mycelium was collected by suction filtration and immediately flash frozen in liquid nitrogen. Total RNA was extracted using RNAiso™ reagent (TaKaRa, Tokyo, Japan) from mycelia ground to fine powders in a frozen mortar and pestle. The manufacturer’ instructions were followed for RNA purification. PrimeScript RT reagent kit (Takara, Tokyo, Japan) was used to reversely transcribe the total RNA into cDNA. In this article, an optimized qPCR system containing 1 μL cDNA template (10-fold diluted), 0.2 μM each primer and 1 × SYBR Premix Ex Taq™ in a final volume of 10 μL was applied. Real-time reverse transcription PCR was performed using the Roche LigntCycler 480 II system. Data were analyzed using the relative quantitation (2^−ΔCt^) method with an endogenous gene *actin* as the internal reference gene [36]. Three biological replicates were carried out for each experiment. The sequences of the primers used for qPCR are shown in Appendix A.

## 3. Results

### 3.1. The ERAD-Deficient Mutants Displayed Severe Growth Defects in Response to the Increased ER Stress

The ERAD pathway maintains the ER homeostasis through degrading misfolded proteins so as to ensure the normal growth and physiological state of cells [37]. To investigate the effect of this pathway on the growth of the important industrial filamentous fungus *T. reesei*, three crucial components of ERAD (Hrd1, Hrd3 and Der1) were knocked out, and the *hrd1*/*hrd3*/*der1* deletion strains were constructed (Appendix A). Then, the mutant strains were grown on the MM plates with different carbon sources, including glucose, lactose, glycerol, and the PDA plate. and no prominent discrepancy in growth rate was discovered compared to the parental strain QM9414 (Appendix A). Afterward, the mutant strains were grown under ER stress, where the MM plates were supplemented with different concentrations from 0 to 15 mM of dithiotreitol (DTT), a reducing agent that can lead to protein misfolding. Meanwhile, the parental strain QM9414 treated with the proteasome inhibitor MG132, which can induce severe ER stress and autophagy by inhibiting proteasome activity and protein degradation, was used as the positive control [38]. As shown in Figure 1, the mutants showed indistinguishable morphology from the parental strain QM9414 at 5 mM DTT, but exhibited a reduction in radial growth and aerial hyphal masses at 10 mM DTT. When the DTT concentration was increased to 15 mM, the Δ*der1* strain could grow slightly, while the Δ*hrd1* and Δ*hrd3* strains could no longer grow, which resembled the strains treated with MG132 (Figure 1A,B). These results indicated that deletion of the key ERAD elements, especially Hrd1 and Hrd3, severely affect the growth and the ER stress resistance of *T. reesei*.

### 3.2. Cell Wall Stability of the ERAD-Deficient Mutants Was Decreased under ER Pressure

The ER pressure has a great impact on maintaining cell wall integrity [39,40]. It is known that CFW (Calcofluor white) or CR (Congo red) can induce cell wall stress and thus can be used to detect the cell wall stability in fungi [32]. Here, the mutant strains were inoculated into the MM plates containing CFW or CR at concentrations of 0, 20, 40, and 80 μg/mL. In total, 10 mM DTT treatment was applied simultaneously to induce a certain degree of ER pressure. Without DTT treatment, the mutant strains grew consistently with the parental strain, even in the presence of high concentrations of CFW or CR (Appendix A). As shown in Figure 2A,B, under 10 mM DTT treatment, the Δ*der1* strain grew at a similar rate to the parental strain QM9414, while the mycelial growth of Δ*hrd1* and Δ*hrd3* was severely affected when treated with 80 μg/mL CFW or CR. These results suggest that Hrd1 and Hrd3 may be involved in maintaining the integrity of the cell wall.

### 3.3. Thermal Tolerance of the ERAD-Deficient Mutants Was Reduced at High Temperature

Temperature has a significant effect on protein folding. It was reported that heat stress could cause the aggregation of misfolded or unfolded proteins in the ER, and thus, cell growth could be impaired [25,41]. To investigate whether Hrd1, Hrd3 and Der1 support the thermal tolerance of *T. reesei*, the growth phenotypes of the Δ*hrd1*, Δ*hrd3* and Δ*der1* strains were monitored at the temperatures of 25, 30, and 37 °C for 3 days (Figure 3A,B). All strains exhibited restricted growth when cultured at 25 °C and grew routinely when cultured at 30 °C. At 37 °C, the growth of the Δ*hrd1* strain was consistent with that of the parental strain QM9414, but the Δ*hrd3* and Δ*der1* strains showed significantly slower growth. These results indicated that Hrd3 and Der1 probably contribute to the maintenance of heat tolerance in *T. reesei*.

### 3.4. The ERAD Pathway Deficiency Affected the Secretion of Cellobiohydrolase and β-Glucosidase

To examine the effect of ERAD deficiency on the protein secretory pathway, cellulase production was used as an index to evaluate the ability of the fungal cells to secrete proteins. The mutant strains were subjected to fermentation under cellulose conditions, and then, the fermentation broths were taken for analyzing the secreted endoglucanase activity, cellobiohydrolase activity and β-glucosidase activity. Compared with the parental strain QM9414, the Δ*hrd1*, Δ*hrd3* and Δ*der1* strains had no significant changes in their extracellular endoglucanase activity (Figure 4A). However, the activities of cellobiohydrolase and β-glucosidase of the mutant strains were decreased (Figure 4B,C). Especially on the 7th day, the β-glucosidase activity of the Δ*hrd1*, Δ*hrd3* and Δ*der1* mutants was only 32.8%, 26.3% and 30.9% of that of the parental strain, respectively. Subsequently, the influences of the *hrd1*/*hrd3*/*der1* knockout on the secretion of β-glucosidase were further assessed using the esculin plates (Figure 4D). It was discovered that the mutant strains showed a much smaller black zone around the colony than the parental strain QM9414, indicating that the β-glucosidase secretion ability of the mutants was significantly reduced. These results demonstrated that the absence of the ERAD components would result in a reduction of the ability of *T. reesei* to secrete cellobiohydrolase and β-glucosidase.

### 3.5. The ERAD Pathway Deficiency Triggered the Pressure Response in the ER

It is known that the lack of ERAD components would cause a failure to degrade the misfolded proteins in the ER [37]. However, the ERAD-deficient mutants of *T. reesei* exhibited similar growth to the parental strain, indicating that it did not affect normal growth (Appendix A). Thus, does the deficient ERAD pathway trigger an ER pressure response? With regard to this hypothesis, the transcription levels of the UPR-related genes *bip1* and *pdi1* were firstly determined under normal growth conditions using glucose as the carbon source. It was found that the ERAD-deficient mutants exhibited the increased transcription levels of *bip1* and *pdi1* (Appendix A). Moreover, the transcription levels of the *hrd1*, *hrd3* and *der1* genes were also determined (Appendix A), and the results showed that the loss of any one of the three ERAD genes would lead to an increase in the transcription levels of the other two ERAD genes. Subsequently, the ERAD-deficient mutants were cultivated under cellulase production conditions using cellulose as the carbon source and subjected to the ER pressure analysis (Figure 5). It was found that the transcription levels of *bip1* and *pdi1* in the mutants were both distinctly enhanced, indicating that the strong UPR was activated (Figure 5A). Especially, the Δ*hrd1* strain exhibited the highest transcription levels of the *bip1* and *pdi1* genes on the 7th day of fermentation, which were 9.2- and 5.3-fold higher than the parental strain, respectively. Meanwhile, the effect of the *hrd1*/*hrd3*/*der1* gene knockout on the ERAD pathway itself was also detected (Figure 5B). The transcription levels of *hrd3* and *der1* in Δ*hrd1*, *hrd3* and *der1* in Δ*hrd1*, as well as *hrd3* and *der1* in Δ*hrd1*, were significantly increased after five days and reached the peak on the 7th day. Particularly, the transcription levels of *hrd3* in Δ*hrd1* on the 7th day of fermentation had the highest increase, reaching 6.3 times higher. Taken together, these results showed that the ERAD pathway deficiency activates the ER stress, especially under the condition of mass secretion of extracellular proteins such as cellulases.

### 3.6. The ERAD Pathway Deficiency Resulted in Downregulating the Transcription of the β-Glucosidase-Coding Gene bgl1

It is reported that the genes encoding extracellular proteins can be repressed via a feedback mechanism under the condition of the ER stress in response to the damage of protein folding or transport, called repression under secretion stress (RESS) [30]. Therefore, it was reasonably assumed that the reduced cellulase production in the ERAD-deficient mutants was probably due to the activation of RESS under ER stress. Here, the transcription levels of the main cellulase genes (*eg1*, *eg2*, *cbh1* and *bgl1*) in the ERAD-deficient mutants were detected (Figure 6). It was found that there were no prominent changes in the transcription levels of the endoglucanase-coding genes *eg1*, *eg2*, and the cellobiohydrolase-coding gene *cbh1* between the mutants and the parental strain QM9414 (Figure 6A–C). However, the transcription levels of the β-glucosidase-coding gene *bgl1* of the Δ*hrd1*, Δ*hrd3* and Δ*der1* strains displayed a remarkable decrease, which was only 35.27%, 16.84% and 15.56% of that of the parental strain QM9414 on the 7th day of fermentation, respectively (Figure 6D). This phenomenon was possibly due to activation of the RESS pathway that downregulated *bgl1* transcription in the mutants, thus resulting in decreased β-glucosidase activity as shown in Figure 4C. Taken together, the ERAD pathway deficiency exerted different influences on secreted proteins, especially β-glucosidase, in *T. reesei*.

## 4. Discussion

*T. reesei* has long been one of the most powerful industrial fungi for cellulase production. In pursuit of greater cellulase production for industrial use, different molecular mechanisms involved in gene transcription regulation and protein secretion have been investigated in *T. reesei* [5,42,43]. Recently, it has been found that the ER secretion pathway plays an important role in the production of cellulases [7,44]. However, the relevance of the ER-mediated protein degradation (ERAD) in cellulase secretion remains poorly understood. In this study, we showed that the ERAD pathway deficiency impacted fungal growth and cellulase secretion.

The main role of the ERAD pathway is to degrade unfolded or misfolded secreted proteins accumulated in the ER, which requires the involvement of the HRD1 complex [16]. The complex is composed of the ubiquitin ligase Hrd1 and three other membrane proteins (Hrd3, Der1 and Usa1). Among the components, Hrd1, Hrd3 and Der1 play an essential role in substrate recognition and delivery and thus were selected as targets to investigate the possible role of the ERAD pathway in fungal growth and cellulase secretion.

The fungal growth of the ERAD-deficient mutants was firstly explored under the normal growth conditions with different carbon sources and the ER stress conditions treated with DTT. The results showed that deletion of the ERAD gene *hrd1*/*hrd3*/*der1* did not affect the normal growth and carbon source utilization of *T. reesei* (Appendix A), which was similar to what has been described in *S. cerevisiae*, *A. fumigatus* and *A. niger* [25,40,45]. It was speculated that it might be due to the upregulation of UPR to make up for the defects of ERAD in order to maintain ER homeostasis (Appendix A). The Δ*derA*Δ*hacA* strain with severely damaged growth in *A. fumigatus* supports this notion [46]. DTT is a commonly used reagent to induce ER stress, and 10 mM DTT is usually used to induce ER stress in *T. reesei* [30,47]. In order to analyze changes in the ER pressure, 0–15 mM concentration gradient of DTT was used in this study. The fungal growth of the Δ*hrd1* and Δ*hrd3* strains was seriously damaged under the ER stress caused by DTT concentrations greater than 10 mM, demonstrating that the mutants have increased sensitivity to the ER stress (Figure 1). Furthermore, DTT was used in combination with CFW/CR to further detect the cell wall stability of the ERAD-deficient mutants. CFW and CR are currently known to induce cell wall stress in fungi [40]. It was found that loss of *hrd1*/*hrd3* under the DTT-induced ER stress and the CFW/CR-induced cell wall stress resulted in impaired growth, suggesting reduced cell wall stability in the Δ*hrd1* and Δ*hrd3* strains (Figure 2). HrdA and DerA act synergistically to support the growth of *A. fumigatus* under cell wall stress conditions [40]. In terms of heat resistance, it was found that Hrd3 and Der1 are involved in the thermotolerance of *T. reesei*, and Hrd1 may not have this function (Figure 3). However, it has been reported that the Δ*hrdA* mutant of *A. fumigatus* has increased thermosensitivity, and Δ*derA* shows no affected thermal stability [40]. In addition, the heat stress response in *Arabidopsis* revealed that Hrd1 could mediate negative regulation of heat tolerance [48]. This may be due to the functional differentiation of Hrd1/HrdA or Der1/DerA among different species. Taken together, we found that deletion of key ERAD components in *T. reesei* had different effects on ER stress sensitivity, cell wall stability and thermal stability, possibly indicating the high sensitivity of *T. reesei* to various stresses.

*T. reesei* is a cell factory with the powerful capacity to produce cellulases. Potential bottlenecks in cellulase production have been evaluated over the past decades, and protein degradation is considered to be related to cellulase production. Here, the cellulase secretion in the ERAD-deficient mutants was investigated under the cellulase-inducing conditions using MCC as the carbon source. The endoglucanase production in the ERAD-deficient mutants was basically consistent with the parental strain QM9414 (Figure 4A,B), while the production of cellobiohydrolase and β-glucosidase in the ERAD-deficient mutants was decreased in the late stage of fermentation (Figure 4B,C). The influences of ERAD pathway deficiency on the proteins beyond cellulases were not found. The effects of the ERAD pathway deficiency on the secretion of cellulases are different, which may be related to the properties of the protein itself, or to the recognition of substrates by ERAD.

There are two major pathways that regulate ER stress and maintain ER homeostasis in eukaryotic cells, namely UPR and ERAD [49]. Therefore, the state of ER stress is mainly manifested by the degree of UPR and ERAD. The transcription levels of the *bip1* and *pdi1* genes were used to measure the UPR degree, and those of the *hrd1*, *hrd3*, and *der1* genes could represent the ERAD degree [50]. Here, the transcription levels of *bip1* and *pdi1* in the ERAD-deficient mutants were significantly elevated under cellulase-inducing conditions (Figure 5). Increased levels of *bipA* mRNA relative to the wild type were also found in the Δ*hrdA* mutant of *A. fumigatus* [37]. Meanwhile, deletion of the ERAD gene (*hrd1/hrd3/der1*) under the cellulase-inducing condition also significantly enhanced the transcription levels of other ERAD components (Figure 5). In particular, deletion of *hrd1* not only led to a significant increase in the transcriptional levels of *bip1* and *pdi1*, but it also had a greater impact on the transcriptional enhancement of other ERAD components than other mutants. This is probably due to the fact that Hrd1 is a multi-transmembrane E3 ubiquitin ligase, playing a key role in the Hrd1 complex, which is essential for the ERAD-L pathway [51]. Taken together, deletion of key ERAD genes leads to ER stress during cellulase secretion, activates the UPR and ERAD pathway to cope with the ER stress, and maintains ER homeostasis. It has been reported that ER stress could lead to repression under secretory stress (RESS), a feedback mechanism that selectively reduces transcript levels of genes encoding endogenous secreted proteins under ER stress, resulting in reduced levels of protein secretion [30]. This secretory stress response caused by impaired protein secretion has been discovered in both *A. niger* and *T. reesei* [30,52]. Here, it was found that the transcription level of *bgl1* was significantly decreased, which probably explains the decreased β-glucosidase production caused by RESS (Figure 6). In addition, the RESS pathway may act by inhibiting the promoter region of genes. It has been reported that the RESS pathway may be involved in transcriptional regulation of the cis-element “TCACGGGC” motif in the *amyB* promoter in *A. oryzae* [53]. However, the promoter sequence of the *bgl1* gene with the significantly downregulated transcription levels in the ERAD-defective mutants was analyzed and found to have no similar binding motif like it. On the other hand, although the transcription level of *cbh1* was not affected, the activity of the secreted cellobiohydrolase was reduced (Figure 6). It is likely that under the pressure of protein secretion, the residual ERAD function may selectively degrade cellobiohydrolase, which accounts for the largest proportion of *T. reesei* extracellular cellulases [3,33].

In conclusion, we investigated the function of key ERAD genes (*hrd1*, *hrd3* and *der1*) in fungal growth and cellulase secretion of *T. reesei*. It was found that *hrd1*, *hrd3* and *der1* played different roles in maintaining the ER homeostasis, cell wall stability and thermal resistance of *T. reesei*. Moreover, we summarized the possible regulation mechanism of cellulase secretion in the *T. reesei* ERAD-deficient mutants under cellulase-inducing conditions (Figure 7). The newly synthesized cellulases should exit the ER after being properly folded by post-translational modifications. However, in the ERAD-deficient mutants, a large number of unfolded or misfolded cellulases can accumulate in the ER, causing ER stress. At this time, the fungal cells respond to the stress through multiple pathways, including the UPR response, which can increase the expression of molecular chaperones, such as Bip1, and foldases, such as Pdi1, to promote protein folding; the ERAD pathway, which can degrade misfolded cellulase to maintain ER homeostasis; and the RESS pathway, which can cause the downregulation of cellulase genes to reduce the ER pressure. The effects of the ERAD pathway deficiency on the secretion of different cellulases varied, which was specifically reflected in that the secretion of endoglucanase was not affected, and the secretion of cellobiohydrolase and β-glucosidase was limited. It was further discovered that activation of the RESS pathway may lead to the reduced transcription of *bgl1*, thus causing a decrease in β-glucosidase production. Taken together, our study demonstrates that the intact function of the ERAD pathway is important for maintaining fungal growth under stresses and normal secretion of cellulases, which sheds light on the mechanism of protein secretion in *T. reesei*.

## Figures and Tables

**Figure 1 jof-09-00074-f001:**
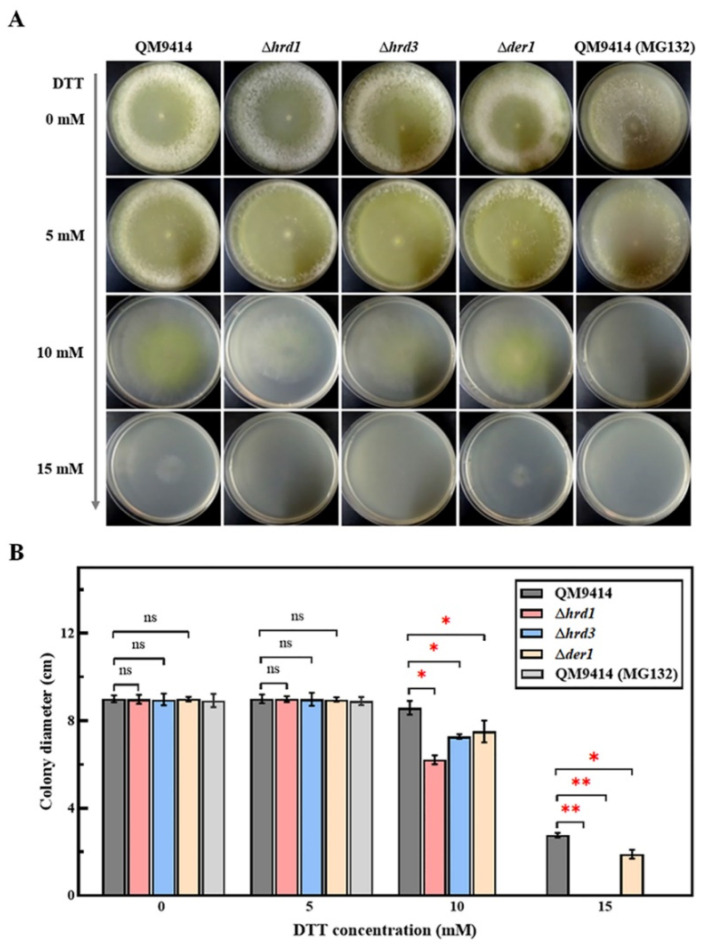
Colony morphology and growth of the ERAD-deficient mutants under different concentrations of the stress agent DTT. (**A**) Plate photographs show the morphology of the control strains and the genetically manipulated ERAD-deficient mutants with increasing concentrations of DTT. The DTT concentrations added to the medium were 0, 5, 10 and 15 mM, respectively. The parental strain QM9414 treated with the proteasome inhibitor MG132 was used as the control. The strains were grown on MM containing glucose as the carbon source at 30 °C for 3 days. (**B**) Colony diameter of QM9414 (treated without/with MG132) and the ERAD-deficient mutants with increasing concentrations of DTT. Values represent the mean of three repeated measurements taken from at least three parallel experiments. The error bars refer to the standard deviations. The difference between the parental strain and the knockout strains was shown by analysis of ANOVA followed by Tukey’s test. * *p* < 0.05; ** *p* < 0.01. ns, not significant.

**Figure 2 jof-09-00074-f002:**
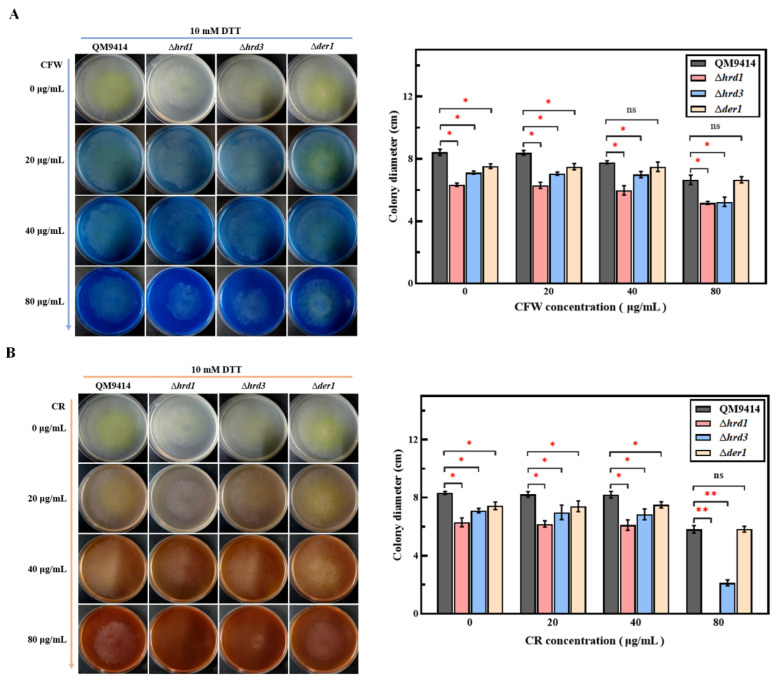
Cell wall stability of the ERAD-deficient mutants under ER pressure. Growth analysis of QM9414 and the ERAD-deficient mutants using the MM plates supplemented with different concentrations of Calcofluor white (**A**) or Congo red (**B**). The concentrations of CFW or CR were 0, 20, 40, and 80 μg/mL, respectively. All plates contained 10 mM DTT to induce ER pressure. The strains were grown on MM containing glucose as the carbon source at 30 °C for 3 days, and the diameters of all colonies were measured. Values represent the mean of three repeated measurements taken from at least three parallel experiments. The error bars refer to the standard deviations. The difference between the parental strain and the knockout strains was shown by analysis of ANOVA followed by Tukey’s test. * *p* < 0.05; ** *p* < 0.01. ns, not significant.

**Figure 3 jof-09-00074-f003:**
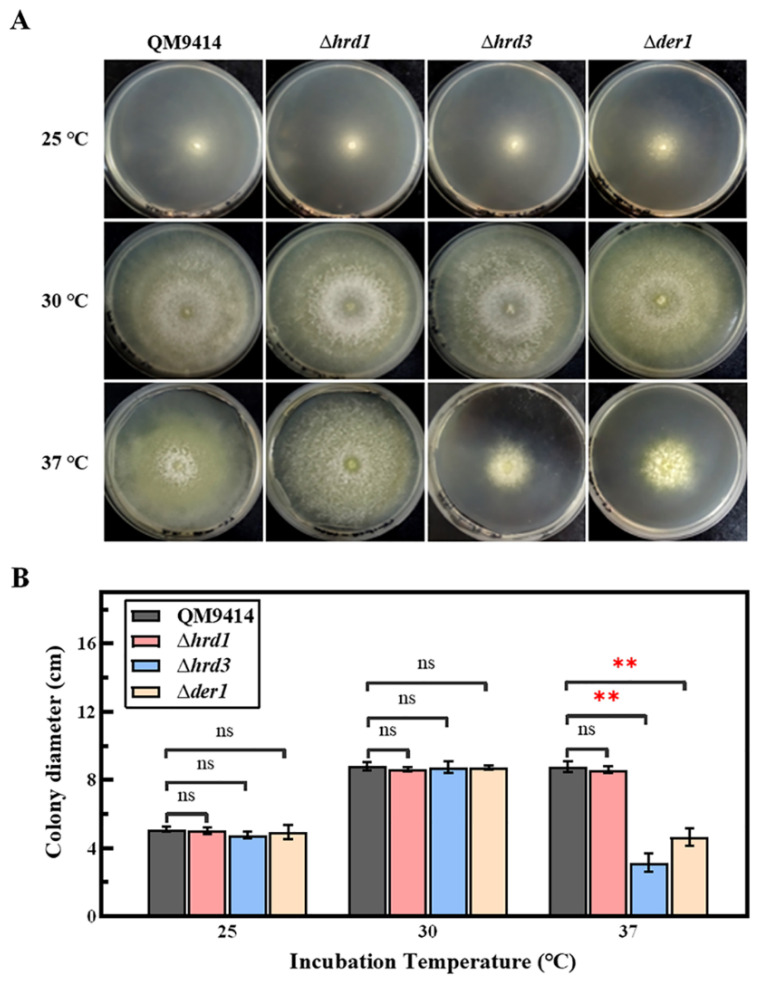
Thermal tolerance of the ERAD-deficient mutants at different culture temperatures. (**A**) The parental strain QM9414 and the ERAD-deficient mutants were grown on MM containing glucose as the carbon source and incubated under the given temperature conditions (25, 30, 37 °C). Growth was monitored for 3 days. (**B**) The colony diameters of QM9414 and the ERAD-deficient mutants were measured at different culture temperatures. Values represent the mean of three repeated measurements taken from at least three parallel experiments. The error bars refer to the standard deviations. The difference between the parental strain and the knockout strains was shown by analysis of ANOVA followed by Tukey’s test. ** *p* < 0.01. ns, not significant.

**Figure 4 jof-09-00074-f004:**
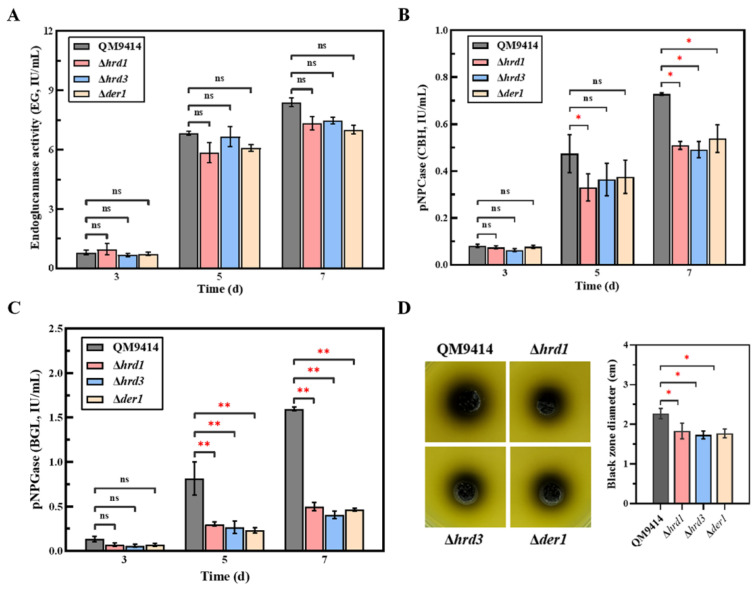
Cellulase secretion of the ERAD-deficient mutants under cellulase induction conditions. The endo-β-1,4-glucanase (EG) activity (**A**), the cellobiohydrolase (CBH) activity (**B**) and the β-glucosidase (BGL) activity (**C**) were detected in fermentation supernatants on the 3rd, 5th and 7th days. The parental strain QM9414 and the ERAD-deficient mutants were cultivated in a cellulase-inducing medium containing 2% microcrystalline cellulose and 2% corn steep liquor at 30 °C. (**D**) Intuitive analysis of the secreted β-glucosidase activity in the CMC-esculin plates experiment. The secretion of β-glucosidase was evaluated by measuring the diameter of the resulting black zone. Values represent the mean of three repeated measurements taken from at least three parallel experiments. The error bars refer to the standard deviations. The difference between the parental strain and the knockout strains was shown by analysis of ANOVA followed by Tukey’s test. * *p* < 0.05; ** *p* < 0.01. ns, not significant.

**Figure 5 jof-09-00074-f005:**
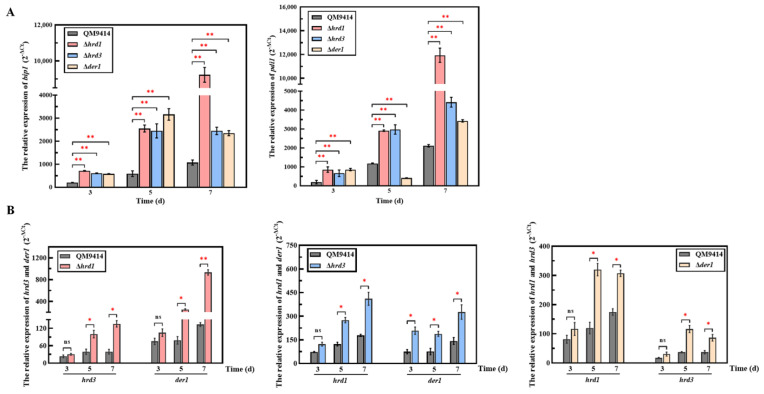
The ER pressure of the ERAD-deficient mutants under cellulase-induced secretion conditions. (**A**) The relative transcription levels of UPR-related genes *bip1* and *pdi1* in the ERAD-deficient mutants under cellulase-induced secretion conditions. (**B**) The relative transcription levels of *hrd1*/*hrd3*/*der1* in the ERAD-deficient mutants under cellulase-induced secretion conditions. Cellulase induction medium contained 2% microcrystalline cellulose and 2% corn steep liquor, which promotes the production of large amounts of cellulase. The relative transcription level of each gene was calculated by the qPCR results of the fermentation samples on the 3rd, 5th, and 7th days. *Actin* gene was universally used to normalize gene transcription levels. Values represent the mean of three repeated measurements taken from at least three parallel experiments. The error bars refer to the standard deviations. The difference between the parental strain and the knockout strains was shown by analysis of ANOVA followed by Tukey’s test. * *p* < 0.05; ** *p* < 0.01. ns, not significant.

**Figure 6 jof-09-00074-f006:**
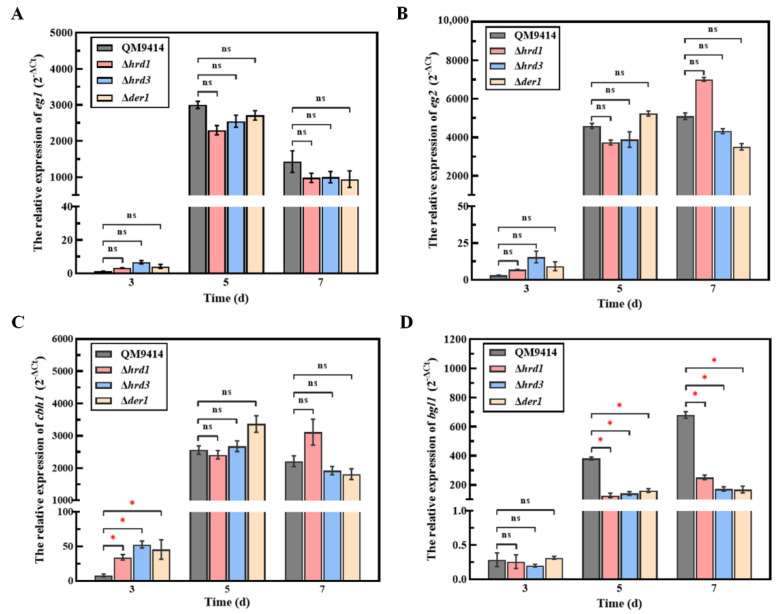
The transcription levels of cellulase component genes in ERAD-deficient mutants under cellulase-induced secretion conditions. (**A**) The relative transcription level of endoglucanase I gene *eg1* in the ERAD-deficient mutants under cellulase-induced secretion conditions. (**B**) The relative transcription level of endoglucanase II gene *eg2* in the ERAD-deficient mutants under cellulase-induced secretion conditions. (**C**) The relative transcription level of cellobiohydrolase I gene *cbh1* in the ERAD-deficient mutants under cellulase-induced secretion conditions. (**D**) The relative transcription level of β-glucosidase gene *bgl1* in the ERAD-deficient mutants under cellulase-induced secretion conditions. Cellulase induction medium contained 2% microcrystalline cellulose and 2% corn steep liquor, which promoted the production of large amounts of cellulase. The relative transcription level of each gene was calculated by the qPCR results of the fermentation samples on the 3rd, 5th, and 7th days. *Actin* gene was universally used to normalize gene transcription levels. Values represent the mean of three repeated measurements taken from at least three parallel experiments. The error bars refer to the standard deviations. The difference between the parental strain and the knockout strains was shown by analysis of ANOVA followed by Tukey’s test. * *p* < 0.05. ns, not significant.

**Figure 7 jof-09-00074-f007:**
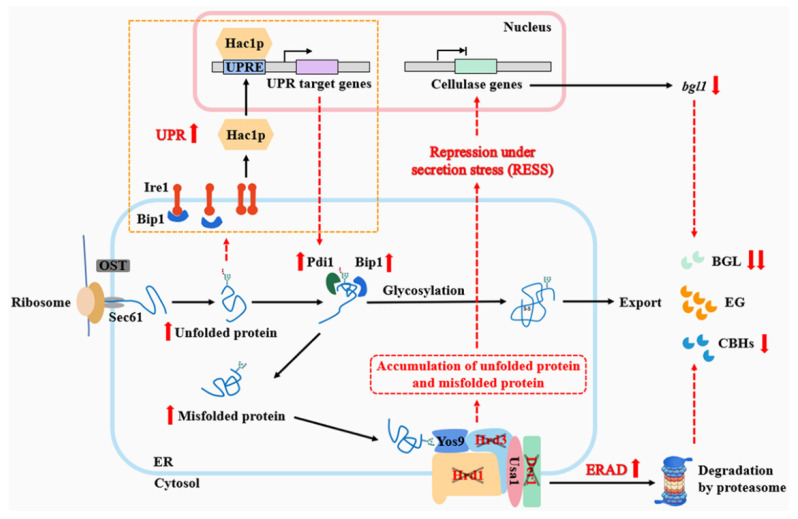
The mechanism diagram of cellulase secretion changes caused by the deletion of critical single ERAD components under high cellulase secretion conditions in *T. reesei*. When the newly synthesized cellulase is translocated across the ER membrane co-translationally via the ribosome-sec61 translocation machinery, it undergoes a series of protein folding and post-translational modifications, such as molecular chaperone binding, disulfide bond formation and glycosylation, etc. Properly folded cellulase exits the ER, while unfolded or misfolded cellulase is recognized by Yos9 and directed to the ERAD pathway for degradation, where it is dislocated from the ER mediated by the Hrd1 ubiquitin ligase complex (Hrd1, Hrd3, Der1) and degraded by the cytosolic ubiquitin–proteasome system. Furthermore, when the baroreceptor Ire1 on the ER membrane senses a large number of unfolded and misfolded proteins, the UPR pathway, including Bip dissociation, Ire1 dimer phosphorylation, Hac1 translation and binding to the UPRE region, is initiated and upregulates molecular chaperones and folding enzymes expression, such as Bip1 and Pdi1. Under the ER pressure caused by the single-component deletion of the Hrd1 complex (Hrd1/Hrd3/Der1), unfolded and misfolded cellulases accumulate, which strongly activates the UPR, ERAD and RESS pathways. ERAD single-component deletion strengthens other components to make up for its absence, possibly resulting in degradation of cellobiohydrolase and decreased secretion. At this time, the ER stress prompts the cell to activate the RESS pathway, which may significantly reduce the expression and secretion of β-glucosidase. Gray cross means that the ERAD component (Hrd1/Hrd3/Der1) has been knocked out. Red dotted line represents the impact caused by ERAD single-component deletion. Red up and down arrows respectively indicate the increase and decrease in elements after ERAD single-component deletion.

## Data Availability

Not applicable.

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
