# Peer review of "The ERAD Pathway Participates in Fungal Growth and Cellulase Secretion in Trichoderma reesei"

_jof, 2023, doi:10.3390/jof9010074_

Round 1

Reviewer 1 Report

I suggest the resubmission of the article after extensive review mainly of the experimental procedures and additions pointed out in the comments sent to the authors. The article presents a good scientific quality, however, the methodology needs to be improved to guarantee the robustness of the data. Therefore, the article cannot be accepted for publication as it is. I would like to review it after the suggested review.

Author Response

Responses to the review comments of Manuscript jof-2092949 entitled “The ERAD pathway participates in the fungal growth and cellulase secretion in Trichoderma reesei

Dear reviewer and editor,

Many thanks for the evaluation and comments on our manuscript jof-2092949, which definitely improve its quality. We have made good use of these comments and suggestions to clarify the indistinct points and to enhance the manuscript. Now, we have incorporated all comments in the revised manuscript and responded to the queries point by point in the following attachment. And in the revised manuscript, all the revisions are shown in blue highlight.

We hope that the revised manuscript is suitable for publication in Journal of Fungi. If there are any questions, just let us know. Thanks again.

Sincerely yours,

Yaohua Zhong

Ph.D, Professor

State Key Laboratory of Microbial Technology, Institute of Microbial Technology

Shandong University, Qingdao 266237, P. R. China

Tel +86 531 88366118 (office); +86 13853149665 (mobile)

Reviewer 2 Report

In this manuscript, hrd1, hrd3 and der1 which were involved in the critical parts of the ERAD pathway of Trichoderma reesei were disrupting. The mycelium growth of the ERAD-deficient mutants on different sources, or under ER stress or cell wall stress was analyzed. The transcription and secretion of cellobiohydrolase and β-glucosidase of above mutants were analyzed. This study indicated the ERAD pathway of T. reesei was related to the growth under stresses and normal secretion of cellulases. This study is significant for further understanding the synthesis and secretion of cellulase.

The study suggested that the ERAD pathway regulated the secretion of endoglucanase and β-glucosidase, and the ERAD pathway deficiency triggered the pressure response in the ER when glucose or cellulose was used as the carbon source. However, ERAD pathway deficiency did not affect the growth on different carbon sources such as glucose, glycerol, lactose. The mycelium growth of ERAD-deficient mutants when cellulose was used as the carbon source should be study and the pressure response in the ER when cellulose was used as the carbon source also should be analyzed. The existing data seems to suggest that when using cellulose as a carbon source, The ER pressure of T. reesei is higher than that of glucose as carbon source, but the author has not made in-depth analysis. If further research shows that cellulose could trigger the pressure response in the ER, ERAD pathway may response the ER pressure caused by cellulose and affect the expression of cellulase. It is of great significance to understand the physiological functions of ERAD pathway in the synthesis of cellulase by T. reesei.

In this manuscript, the transcription of main cellulase genes were analyzed by QPCR. Only the transcription levels of the bgl1 of the ERAD-deficient mutants displayed a remarkable decrease. The protein level changes of main cellulases in intracellular and extracellular should be analyzed, as well as the changes of their unfolded or misfolded protein levels.

Line 220, “… under the cell wall stress” should be “… under the ER pressure”.

Author Response

(The authors gave the same response as above.)

Round 2

Reviewer 2 Report

After revision, this manuscript is suitable for acceptance and publication in JOF.